

# Identification of key genes in non-small cell lung cancer by bioinformatics analysis

Li Zhang[1], Rui Peng[2], Yan Sun[1], Jia Wang[1], Xinyu Chong[1] and Zheng Zhang[1]

[1] Department of Molecular Medicine and Cancer Research Center, Chongqing Medical University, Chongqing, China

[2] Department of Bioinformatics, Chongqing Medical University, Chongqing, China

## ABSTRACT

**Background.** Non-small cell lung cancer (NSCLC) is one of the most common malignant tumors in the world, and it has become the leading cause of death of malignant tumors. However, its mechanisms are not fully clear. The aim of this study is to investigate the key genes and explore their potential mechanisms involving in NSCLC.

**Methods.** We downloaded gene expression profiles GSE33532, GSE30219 and GSE19804 from the Gene Expression Omnibus (GEO) database and analyzed them by using GEO2R. Gene Ontology and the Kyoto Encyclopedia of Genes and Genomes were used for the functional and pathway enrichment analysis. We constructed the protein-protein interaction (PPI) network by STRING and visualized it by Cytoscape. Further, we performed module analysis and centrality analysis to find the potential key genes. Finally, we carried on survival analysis of key genes by GEPIA.

**Results.** In total, we obtained 685 DEGs. Moreover, GO analysis showed that they were mainly enriched in cell adhesion, proteinaceous extracellular region, heparin binding. KEGG pathway analysis revealed that transcriptional misregulation in cancer, ECM-receptor interaction, cell cycle and p53 signaling pathway were involved in. Furthermore, PPI network was constructed including 249 nodes and 1,027 edges. Additionally, a significant module was found, which included eight candidate genes with high centrality features. Further, among the eight candidate genes, the survival of NSCLC patients with the seven high expression genes were significantly worse, including CDK1, CCNB1, CCNA2, BIRC5, CCNB2, KIAA0101 and MELK. In summary, these identified genes should play an important role in NSCLC, which can provide new insight for NSCLC research.

# INTRODUCTION

Lung cancer is one of the most common malignant tumors in the world, and it has become the first cause of death in malignant tumors in urban population in China (*Yang et al., 2018*). Non-small cell lung cancer (NSCLC) as the main type of lung cancer accounts for about 80% of all cases compared with small cell carcinoma and about 75% of patients

Corresponding author
Zheng Zhang,
zhangzheng92@163.com

are already in the advanced stage (*Boolell et al., 2015*). Although prominent achievements have been made in early prognosis and treatment, the five-year survival rate of NSCLC is not optimistic (30%–40%) (*Spira & Ettinger, 2004*). Thus, it is important to explore the molecular mechanisms of non small cell lung cancer to develop more effective treatment methods.

In recent years, a large number of bioinformatics have been applied to clinical research, and a number of disease-related data have also been produced (*Kulasingam & Diamandis, 2008*). It will provide more significant diagnosis and treatment of NSCLC and find more potential targets for more effective therapeutic strategies. As more and more high throughput sequencing technologies and microarray have been released, there have a wide range of medical oncology research. Recently, microarray technology was used to investigate genes expression profiling which have identified differentially expressed genes (DEGs) in tumor samples compared with non-tumor samples for the development and progression of NSCLC (*Jin et al., 2016*). However, the pathogenesis of NSCLC is inadequately understood.

In this study, we downloaded three original gene expression profile datasets from the Gene Expression Omnibus, including GSE33532 (*Li et al., 2018*), GSE30219 (*Miao et al., 2019*) and GSE19804 (*Ni et al., 2018*). A total of 685 differently expressed genes (DEGs) were identified between NSCLC tissues and the non-tumor tissues using GEO2R. In addition, the Gene Ontology (GO) functional annotation and Kyoto Encyclopedia of Genes and Genomes (KEGG) pathway analysis were performed using the DAVID database. A protein-protein interaction (PPI) network was constructed using the Search Tool for the Retrieval of Interacting Genes (STRING). Moreover, module analysis and centrality analysis of the network were used to identify the potential core genes in NSCLC. Further, prognosis and survival analysis results showed seven key genes were related to the prognosis of NSCLC. The work will provide the help to search for the molecular targets for the diagnosis and treatment of NSCLC.

## MATERIALS AND METHODS

### Microarray data

Three gene expression profiles (GSE33532, GSE30219 and GSE19804) were obtained from the Gene Expression Omnibus database (http: //www.ncbi.nlm.nih.gov/geo/). All three datasets followed these criteria: (a) The samples consisted of two groups of NSCLC tissues and non-tumor tissues; (b) The sample size of each study would be greater than 100; (c) They were updated recently (2018-2019); (d)The samples based on the same platform: GPL570. The array data of GSE33532 included 80 NSCLC interpatient (patient-to-patient) and intrapatient (tumor sub-sample) variations in stage I and II of NSCLC patients in Germany. GSE30219 consisted of 293 NSCLC tissue samples and 14 non-tumor samples in France. GSE19804 included 60 paired NSCLC and non-tumor tissue of non-smoking female lung cancer patients in Taiwan population (*Lu et al., 2010*).

### Identification of DEGs

The DEGs between NSCLC tissues and non-tumor tissues were identified by GEO2R (https://www.ncbi.nlm.nih.gov/geo/geo2r/), which is the Gene Expression Omnibus

online tool to screen DEGs by comparing two different groups of samples under the same experimental condition (Piao et al. 2018). The adjusted $P$-value ($q$-value) was regarded as the standard to correct the occurrence of false-positive results using the Benjamini and Hochberg false discovery rate method. The cut-off criteria was |log$_2$fold-change (FC)| >1 and $q$-value <0.01. The intersections of DEGs from GSE33532, GSE30219 and GSE19804 were obtained by Venny 2.1.0 (http://bioinfogp.cnb.csic.es/tools/venny/) (*Sun et al., 2018*).

## GO function and KEGG pathway enrichment analysis

GO and KEGG is used to understand and simulate higher-order functional behaviors of cells or organisms from the genomic information (*Hao et al., 2018*). GO and KEGG annotation of DEGs were carried out through DAVID database (https://david.ncifcrf.gov/), which is an online program that can provide a comprehensive set of functional annotation tool (*Huang da, Sherman & Lempicki, 2009*). The analysis of Gene Ontology involves three aspects: cell component, molecular function and biological process. Enrichment factor was considered as the cutoff criterion to indicate a statistically significant difference. And the top 25 GO terms and the KEGG pathways were selected.

## Construction of PPI network and Module analysis of PPI network

STRING is online tool designed to evaluate the PPI information. To detect the potential relationship among those DEGs, we used STRING app in Cytoscape (http://www.cytoscape.org/) and mapped the DEGs into STRING (*Szklarczyk et al., 2019*). The combined score of >0.4 was used as the cut-off value in the STRING database to improve the result (*Shannon et al., 2003*). The Cytoscape app **M**olecular **Co**mplex **D**etection (MCODE) was applied to create the modules in the PPI network (*Bader & Hogue, 2003*), and degree cut-off = 2, node score cut-off = 0.2, k-core = 2, and max depth = 100 were regarded as the criteria. The pathway analysis of genes in each module was performed by DAVID. And GO and KEGG pathway analysis were also made to explore the potential information of the genes.

## Centrality analysis of PPI network

We search and predict the key genes of the network using the significant parameters of degree centrality, betweenness centrality, closeness centrality and eigenvector centrality (*Ostovari, Yu & Steele-Morris, 2018*). The four centrality scores of each vertex were calculated by Cytoscape. Degree, betweenness and closeness were calculated using Network Analyzer of Cytoscape (*Assenov et al., 2008*) and eigenvector was calculated using Cytoscape app CytoNCA (*Tang et al., 2015*). The score file for these four parameters was downloaded from the Cytoscape software, the R language was used to describe the distribution of the four parameters and calculate the correlation among the four key centralities.

## The effect of expressions of key genes on NSCLC patient survival

With the widespread use of chips and high-throughput sequencing, a large amount of genomics data has accumulated in the field of cancer research, such as TCGA (The Cancer Genome Atlas) and ICGC (International Cancer Genome Consortium). Currently, GEPIA (http://gepia.cancer-pku.cn/) is an interactive web server for cancer expression profile
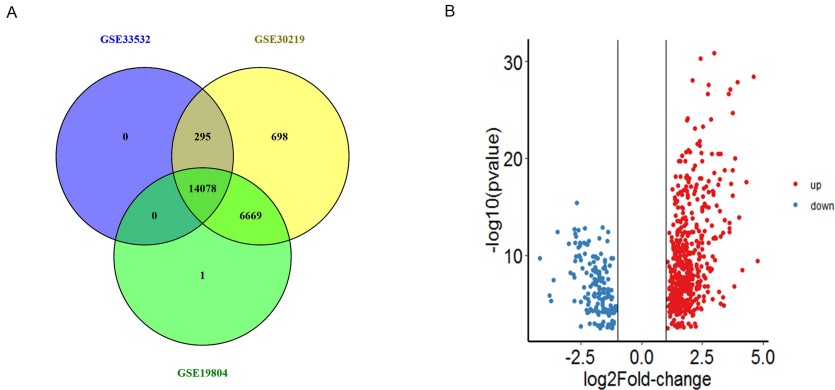

**Figure 1** **The identification of DEGs.** (A) The intersection of DEGs from the expression profiles GSE33532, GSE30219 and GSE19804 were detected by Venny 2.1. (B) Volcano plot of the distribution of all differentially expressed genes, including the 519 upregulated genes (red) and 166 downregulated genes (blue). The cut-off criteria: |logFC| > 1, $p$ value < 0.01.

data containing 9,736 tumor samples and 8,587 normal samples from TCGA (The cancer Genome Atlas) and GTEx (Genotype-Tissue Expression), which provides customizable functions such as tumor and normal differential expression analysis, and we can get the expression of hub genes in NSCLC tissues and normal ones. $P < 0.01$ was selected as a threshold (*Tang et al., 2018a*; *Tang et al., 2018b*).

## RESULT

### Identification of DEGs

In total, we obtained 14373, 21740 and 20748 DEGs from GSE33532, GSE30219 and GSE19804 datasets, 14078 genes were found to be existed in three databases (Fig. 1A and Table S1). Among these genes, we obtained 685 DEGs including 519 were up-regulated and 166 were down-regulated genes between NSCLC tissues and non-tumor tissues (|logFC|>1, $P < 0.01$) (Fig. 1B).

### Go and KEGG pathway enrichment analysis

The online tool of DAVID was used to analyze the GO and KEGG pathway enrichment based upon DEGs. The result of GO showed that the genes were involved in protein binding, integral component of membrane, integral component of plasma membrane, extracellular exosome, extracellular region, extracellular space of the cell component and positive regulation of transcription from RNA polymerase II promoter, cell adhesion, identical protein binding of the molecular function (Fig. 2A). Moreover, the result of KEGG showed that the genes were mainly enriched in transcriptional misregulation in cancer, ECM-receptor interaction, cell cycle and p53 signaling pathway (Fig. 2B).

### Construction of the NSCLC related PPI network

The PPI network of 685 DEGs was constructed from STRING to predict the interactions of identified DEGs, consisting of 249 nodes and 1,027 edges, as shown in Fig. S1. The majority

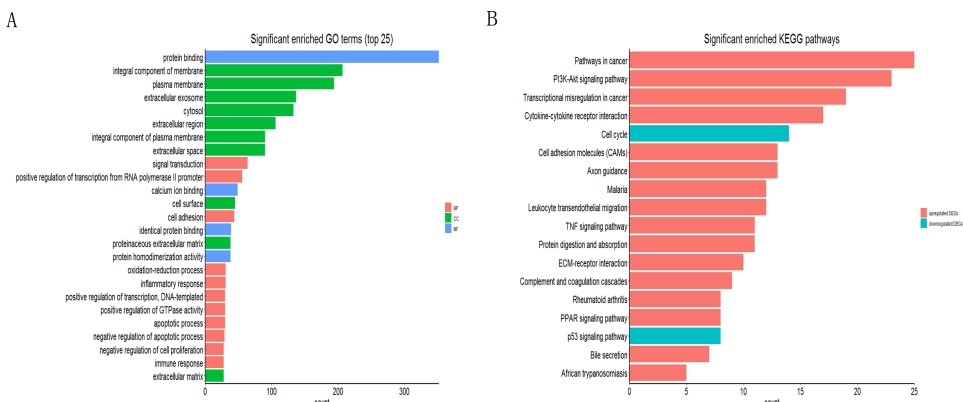

**Figure 2 Functional annotation of DEGs by DAVID.** (A) GO analysis of DEGs based on their functions into three groups (molecular function, biological process and cellular component). Data showed the top 25 significant terms according to enrichment factor. (B) The KEGG pathway analysis of DEGs in NSCLC.

of the nodes in the PPI network were the upregulated DEGs in NSCLC samples. The PPI network mainly showed that the interactions between the molecules.

## Module analysis of the NSCLC related network

To explore the significance of the molecules of the NSCLC related PPI network, the Module analysis was performed by MCODE. Our result showed that there were eight modules in the PPI network by module analysis. A significant module with score >2 was found in Fig. S2. The module 1 was the most significant in whole modules, and it was located in the center of the entire PPI network. Moreover, the first-ranked module had the most nodes (33 nodes such as CDK1, CCNB1, CCNB2, CCNA2, KIAA0101, MELK and so on) and edges (510 edges). Furthermore, the genes were related to some enriched GO terms and pathways such as protein-binding, mitotic nuclear division, cell division, and so on (Table S2).

## Centrality analysis of PPI network

To explore the features of the molecules of the NSCLC related PPI network, the centrality analysis was carried on by Cytoscape. Several parameters are used as centrality parameters in the centrality analysis, including degree, betweenness, closeness, eigenvector, information, subgraph, and so on. Among these, degree, betweenness, closeness and eigenvector are established important parameters in the centrality analysis. Degree centrality is the most direct measure of node centrality in network analysis. Betweenness centrality is an indicator of the importance of a node by the number of shortest paths passing through a node. Closeness centrality reflects the proximity of a node to other nodes in the network. Eigenvector centrality has the property that a vertex has a high score if it is connected with many other vertices with high scores. So, four topological features of the network centrality (degree, betweenness, closeness and eigenvector) were analyzed in our study. Our results showed that the distributions of degree, betweenness and eigenvector displayed the power-law distributions, while the distribution of closeness displayed a heavy-tailed distribution
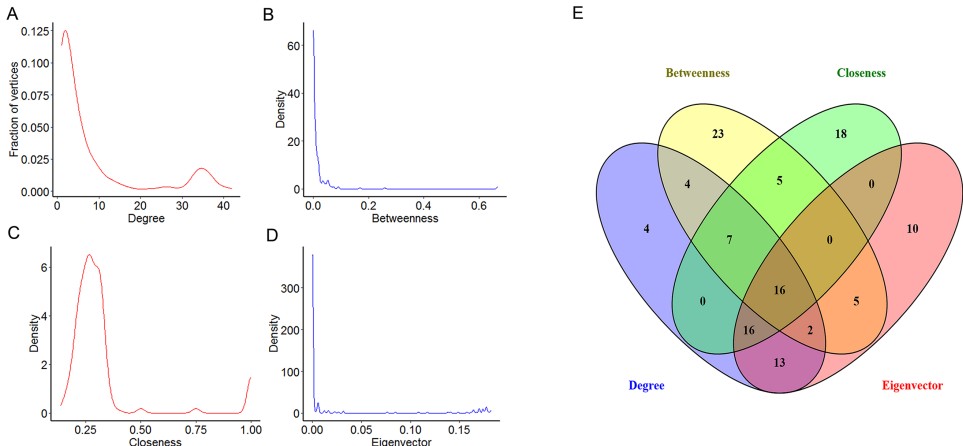

**Figure 3** **The centrality analysis of the NSCLC related PPI network.** (A–D) The distribution of degree centrality, betweenness centrality, closeness centrality and eigenvector centrality based in Cytoscape's plug-in network by the R language. (E) The intersection of the top 25% molecules in each centrality (degree centrality, betweenness centrality, closeness centrality and eigenvector centrality) by Venny 2.1.0. Results showed 16 key DEGs were chosen for further study because of their high degree, betweenness, closeness and eigenvector values.

(Figs. 3A–3D). Moreover, the top 25% of each parameter were chosen for further study (Table 1). Furthermore, 16 key DEGs with high topological features were obtained with high degree, betweenness, closeness and eigenvector (Fig. 3E). Specifically, we found that eight among the 16 key DEGs were also in the module 1, including cyclin dependent kinase 1 (CDK1), cyclin B1 (CCNB1), cyclin A2 (CCNA2), baculoviral IAP repeat containing 5 (BIRC5), cyclin B2 (CCNB2), PCNA clamp associated factor (KIAA0101), maternal embryonic leucine zipper kinase (MELK) and enhancer of zeste 2 polycomb repressive complex 2 subunit (EZH2).

### Survival analysis of key genes in NSCLC
GEPIA was applied to catch the 8 hub genes expression level between NSCLC tissues and normal ones. Our results showed that the seven high expressions of CDK1, CCNB1, CCNA2, BIRC5, CCNB2, KIAA0101 and MELK were associated with worse OS for NSCLC patients ($p < 0.01$) (Figs. 4A–4G).

## DISCUSSION
The occurrence and development of lung cancer is a complex process because it involves aberrations of multiple genes and cellular pathways. It is vital important to find these hub genes and understand their roles in the molecular mechanism of NSCLC to improve the diagnosis and treatment (*Liang, Li & Zhao, 2016*). With the development of microarray and high-throughput technologies, identification of disease-associated genes and genes function prediction were applied.

In this study, based on three expression profiles of GSE33532, GSE30219 and GSE19804, 685 DEGs were identified, consisting of 519 upregulated and 166 downregulated genes

**Table 1** The top 25% centrality parameters of 16 candidate molecules in the PPI network.

| Genes name | Degree | Betweenness | Closeness | Eigenvector |
|---|---|---|---|---|
| CDK1 | 42 | 0.05442045 | 0.34029851 | 0.34029851 |
| PRKCB | 10 | 0.0897726 | 0.35130971 | 0.006357714 |
| EDN1 | 26 | 0.25859441 | 0.40140845 | 0.012624903 |
| TIMP3 | 13 | 0.05680489 | 0.33878158 | 0.005104212 |
| MELK | 36 | 0.01845249 | 0.33628319 | 0.17740285 |
| SPP1 | 13 | 0.03509843 | 0.34131737 | 0.02659089 |
| PLA2G1B | 9 | 0.05464156 | 0.33878158 | 0.006098443 |
| BIRC5 | 37 | 0.0424209 | 0.36018957 | 0.1761744 |
| CCNB1 | 40 | 0.01947945 | 0.32805755 | 0.18236643 |
| KIAA0101 | 37 | 0.0555325 | 0.34285714 | 0.17740497 |
| CCNA2 | 40 | 0.02089539 | 0.32900433 | 0.18205161 |
| EZH2 | 38 | 0.09184724 | 0.36952998 | 0.15754144 |
| CCNB2 | 37 | 0.01609439 | 0.36952998 | 0.15754144 |
| LRRK2 | 22 | 0.16920811 | 0.36190476 | 0.012165083 |
| SELP | 11 | 0.05407089 | 0.34131737 | 0.006136451 |
| KLF4 | 9 | 0.05250658 | 0.34915773 | 0.016391514 |

between NSCLC tissues and non-tumor tissues. Moreover, GO results showed that the genes were involved in cell adhesion, extracellular region, integrin binding. Also, the result of KEGG showed that the genes were mainly enriched in transcriptional misregulation in cancer, ECM-receptor interaction, cell cycle and p53 signaling pathway. As known, the expression of cell adhesion molecules could affect the adhesive and signal transduction status of cells, and intimately related to the processes of cell motility and cell migration (*Han et al., 2017*). Also, researchers had found that integrin binding participated in variety of cellular functions including cell proliferation, survival and differentiation (*Pan et al., 2016*). In addition, cell cycle and cell proliferation were established terms related to tumor proliferation and apoptosis in lung cancer (*An et al., 2016*). Previous studies indicated that ECM-receptor interaction involved in the cell adhesion (*Lessey & Young, 1997*), and ECM played an important role in tissue and organ morphogenesis and in the maintenance of cell and tissue structure and function (*Zakaria et al., 2015*). Moreover, according to previous studies, this cisplatin exerted its cytotoxic effect which associated with the p53 signaling pathway, and the disturbances in the p53 signaling pathway were associated with NSCLC (*Tang et al., 2018a*; *Tang et al., 2018b*). Therefore, the data suggest that the identified DEGs may play roles in the occurrence and development of NSCLC.

To explore the molecular mechanism of NSCLC, the NSCLC related PPI network was constructed. Moreover, Module analysis showed that eight functional modules were detected. Specifically, the first-ranked module was found including 33 nodes and 510 edges. Previous studies have showed that modules analysis have been increasing used for identified the hub genes for various cancers liking hepatocellular carcinoma (*Zhu et al., 2018*), gastric cancer (*Cao et al., 2018*) and small cell lung cancer (*Wen et al., 2018*). Thus, these suggest that the nodes and edges in the first-ranked module may be the significant molecules of

Peer J

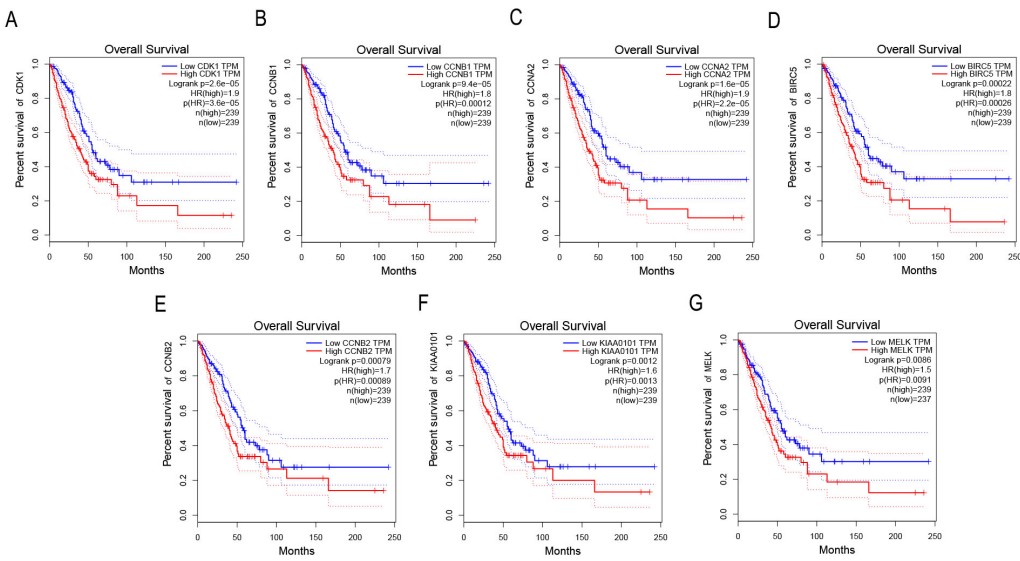

**Figure 4** **Survival analysis of key genes in NSCLC patients.** (A) Survival analysis of CDK1 in NSCLC patients by GEPIA. The result showed the survival of NSCLC patients with the high expressions of CDK1 was significantly worse ($p < 0.01$). (B) Survival analysis of CCNB1 in NSCLC patients by GEPIA. The result showed the survival of NSCLC patients with the high expressions of CCNB1 was significantly worse ($p < 0.01$). (C) Survival analysis of CCNA2 in NSCLC patients by GEPIA. The result showed the survival of NSCLC patients with the high expressions of CCNA2 was significantly worse ($p < 0.01$). (D) Survival analysis of BIRC5 in NSCLC patients by GEPIA. The result showed the survival of NSCLC patients with the high expressions of BIRC5 was significantly worse ($p < 0.01$). (E) Survival analysis of CCNB2 in NSCLC patients by GEPIA. The result showed the survival of NSCLC patients with the high expressions of CCNB2 was significantly worse ($p < 0.01$). (F) Survival analysis of KIAA0101 in NSCLC patients by GEPIA. The result showed the survival of NSCLC patients with the high expressions of KIAA0101 was significantly worse ($p < 0.01$). (G) Survival analysis of MELK in NSCLC patients by GEPIA. The result showed the survival of NSCLC patients with the high expressions of MELK was significantly worse ($p < 0.01$).

the network. Additionally, the results of centrality analysis showed that the distributions of degree, betweenness, closeness and eigenvector were typical form of heavy-tailed or close to typical form of heavy-tailed. As known, the typical forms of heavy-tailed were frequently found in many bio-molecular networks (*Wang, Yu & Lu, 2014*). Therefore, it suggests that the constructed networks have some similar statistical properties as the evolving network. Further, our data found 16 candidate molecules displayed high degree, betweenness, closeness and eigenvector. Finally, we found eight among 16 candidate molecules were consisted of the module 1. Therefore, these results demonstrated that these 8 candidate molecules may be the important molecules in the NSCLC related PPI network and participant in NSCLC.

The survival and prognosis are important analytical indexes for evaluating disease prognosis, especially in cancer research. Our results showed that among the 8 candidate genes, the survival and prognosis of NSCLC patients with the seven high expression genes of CDK1, CCNB1, CCNA2, BIRC5, CCNB2, KIAA0101 and MELK were significantly worse ($p < 0.01$). CDK1, a cyclin-dependent kinase, is the center of the cell cycle control

system and plays a very important role in the cell cycle progression (*Ge et al., 2019*). Studies had shown that CDK1 expression was associated with prognosis in advanced non-small cell lung cancer, and low expression of CDK1 had a better prognosis (*Shi et al., 2017*; *Shi et al., 2016*). BIRC5 is a new member of the apoptosis inhibitory protein family and is the most potent inhibitor of apoptosis. BIRC5 has complex biological functions, inhibits apoptosis, promotes cell transformation and participates in cell mitosis, angiogenesis and drug resistance (*Ebrahimiyan et al., 2019*). It was reported that the expression level of BIRC5 was significantly higher in NSCLC tissues compared to non-tumor tissues, and negatively correlated with Tp53 (*Wang et al., 2018*; *Vayshlya et al., 2008*). CCNB2, a member of the cyclin family proteins, played an important role in the progression of G2/M transition by activating CDK1 kinase, and CCNB2 inhibition induced cell cycle arrest (*Gao et al., 2019*; *Qian et al., 2015*). CCNB1 was found to be associated with tumor aggressiveness and poor survival in lung cancer and esophageal cancer (*Gao & Wang, 2018*; *Soria et al., 2000*). The protein encoded by CCNA2 genes belongs to the highly conserved cyclin family, whose members function as regulators of the cell cycle (*Krautgasser et al., 2019*). However, the results about the function of CCNA2 on the prognosis in NSCLC were mutually contradictory. Some found that the expression of CCNA2 was correlated with prognosis, but others suggested that CCNA2 was not associated with prognosis of NSCLC (*Xiao et al., 2018*; *Ni et al., 2018*). Therefore, the exact function of CCNA2 on prognosis of NSCLC needs further study. KIAA0101 was found to play an important role in the regulation of the cell cycle of NSCLC and the studies have showed that high-level KIAA0101 expression was also identified as an independent prognosis factor in NSCLC (*Li et al., 2010*; *Kato et al., 2012*). MELK, a cell cycle-dependent protein kinase, belongs to the KIN1/PAR-1/MARK family. MELK involved in the process of cell cycle, cell proliferation, tumor formation and apoptosis (*Zhang et al., 2017*). Moreover, MELK expression was elevated and closely related to the prognosis of NSCLC patients (*Mullapudi et al., 2015*; *Giuliano et al., 2018*). Combined these together, It suggests that these seven genes have a close relation with the prognosis of NSCLC patients, and they may act as the prognostic indicators in NSCLC.

## CONCLUSIONS

In conclusion, using multiple profile datasets and integrated bioinformatical analysis, we have identified 685 DEGs candidate genes at screening step, which significant enriched in several pathways, mainly associated with cell cycle and p53 signaling pathway. Then we filtered 249 nodes and 1,027 edges in DEGs protein–protein interaction network complex, and found some candidate genes with high centrality features in a significant module. Finally, 7 key genes were found to be related to the prognosis of NSCLC patients, including CDK1, CCNB1, CCNA2, BIRC5, CCNB2, KIAA0101 and MELK. These findings could significantly improve our understanding of the cause and underlying molecular events in NSCLC, these candidate genes and pathways could be therapeutic targets for NSCLC. When these are combined together, it suggests that these seven genes have a close relation with the prognosis of NSCLC patients, and they may act as the prognostic indicators in NSCLC.

However, the functions and exact mechanisms of these seven genes in the prognosis of NSCLC, cell proliferation, cell migration, and so on need to be further studied.

### Funding

This study was supported by the Ying Yao Scientific Project of Basic Medicine College of Chongqing Medical University, China (JCYY201807). The funders had no role in study design, data collection and analysis, decision to publish, or preparation of the manuscript.

### Competing Interests

The authors declare there are no competing interests.

### Author Contributions

- Li Zhang conceived and designed the experiments, performed the experiments, analyzed the data, contributed reagents/materials/analysis tools, prepared figures and/or tables, authored or reviewed drafts of the paper, approved the final draft.
- Rui Peng conceived and designed the experiments, contributed reagents/materials/analysis tools, authored or reviewed drafts of the paper, approved the final draft.
- Yan Sun performed the experiments, contributed reagents/materials/analysis tools, authored or reviewed drafts of the paper, approved the final draft.
- Jia Wang and Xinyu Chong analyzed the data, contributed reagents/materials/analysis tools, authored or reviewed drafts of the paper, approved the final draft.
- Zheng Zhang conceived and designed the experiments, authored or reviewed drafts of the paper, approved the final draft.

### Microarray Data Deposition

The following information was supplied regarding the deposition of microarray data:

The raw data is available at NCBI GEO: GSE33532, GSE30219, and GSE19804.

GSE33532: https://www.ncbi.nlm.nih.gov/gds/?term=GSE33532.

GSE30219: https://www.ncbi.nlm.nih.gov/geo/query/acc.cgi?acc=GSE30219.

GSE19804: https://www.ncbi.nlm.nih.gov/geo/query/acc.cgi?acc=GSE19804.

### Data Availability

The raw measurements are available in Table S1.

### Supplemental Information

Supplemental information for this article can be found online at http://dx.doi.org/10.7717/peerj.8215#supplemental-information.

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
