# Peer review of "Identification of key genes in non-small cell lung cancer by bioinformatics analysis"

_PeerJ, doi:10.7717/peerj.8215_

## Round 0.1 · original submission · Major Revisions

Please make the corrections suggested by the reviewers.

·

Basic reporting

The article entitles “Identification of target genes in non-small cell lung cancer by bioinformatics analysis “ by Zhang et al is interesting and has potential for publication. However, there are few concerns as listed below and authors need to address these issues properly before the article is accepted.

1) The authors have followed the methods applied in Xiao et al., 2018 (https://www.ncbi.nlm.nih.gov/pmc/articles/PMC5928621/) and Zhang X. 2018. Identification of Candidate Biomarkers Correlated With the Pathogenesis and Prognosis of Non-small Cell Lung Cancer via Integrated Bioinformatics Analysis. Front Genet 9:469. 10.3389/fgene.2018.00469. However, in this article, the method described is unclear, how the tools were used and the detailed description of each analysis need to be provided.

2) In title, authors have mentioned “target genes”. What the target means? Is it drug target/ Marker for prognosis/ anything else please clarify or remove from the title.

3) It is well established that markers are ethnicity specific. Authors have not described the used datasets properly, what the stages of the cancers, what the ethnicity of the patients etc.

4) Further, it is unclear why the authors have selected these particular datasets from GEO, also in the used datasets, the number of controls are less. Therefore, larger and additional datasets to be incorporated in the analysis.

5) It is also unclear about the novelty of the findings. The identified markers are already reported in various literature as following. Therefore, its require to clearly mention the novelty of the work in respect to used method and or outcomes.
CDK1 (https://www.ncbi.nlm.nih.gov/pmc/articles/PMC5356732/)
CCNB1 (https://www.ncbi.nlm.nih.gov/pubmed/10945597)
CCNA2 (https://www.frontiersin.org/articles/10.3389/fgene.2018.00469/full)
BIRC5 (https://link.springer.com/article/10.1134/S0026893308040146)
CCNB2 (https://www.ncbi.nlm.nih.gov/pubmed/26349989)
KIAA0101 (https://www.ncbi.nlm.nih.gov/pubmed/21689861)
MELK (https://elifesciences.org/articles/32838)

6) In conclusion, authors have mentioned “However, further molecular and biological experiments are required to confirm the functions of the key genes in NSCLC.” As mentioned in the above points, all the selected 7 markers are reported previously with validation, what the other experiments the authors propose to conduct to validate? Please elaborate.

Experimental design

1) The authors have followed the methods applied in Xiao et al., 2018 (https://www.ncbi.nlm.nih.gov/pmc/articles/PMC5928621/) and Zhang X. 2018. Identification of Candidate Biomarkers Correlated With the Pathogenesis and Prognosis of Non-small Cell Lung Cancer via Integrated Bioinformatics Analysis. Front Genet 9:469. 10.3389/fgene.2018.00469. However, in this article, the method described is unclear, how the tools were used and the detailed description of each analysis need to be provided

2) It is well established that markers are ethnicity specific. Authors have not described the used datasets properly, what the stages of the cancers, what the ethnicity of the patients etc.

3) It is unclear why the authors have selected these particular datasets from GEO, also in the used datasets, the number of controls are less. Therefore, larger and additional datasets to be incorporated in the analysis.

Validity of the findings

1) It is also unclear about the novelty of the findings. The identified markers are already reported in various literature as following. Therefore, its require to clearly mention the novelty of the work in respect to used method and or outcomes.
CDK1 (https://www.ncbi.nlm.nih.gov/pmc/articles/PMC5356732/)
CCNB1 (https://www.ncbi.nlm.nih.gov/pubmed/10945597)
CCNA2 (https://www.frontiersin.org/articles/10.3389/fgene.2018.00469/full)
BIRC5 (https://link.springer.com/article/10.1134/S0026893308040146)
CCNB2 (https://www.ncbi.nlm.nih.gov/pubmed/26349989)
KIAA0101 (https://www.ncbi.nlm.nih.gov/pubmed/21689861)
MELK (https://elifesciences.org/articles/32838)

2) In conclusion, authors have mentioned “However, further molecular and biological experiments are required to confirm the functions of the key genes in NSCLC.” As mentioned in the above points, all the selected 7 markers are reported previously with validation, what the other experiments the authors propose to conduct to validate? Please elaborate.

Additional comments

Improvement in description of method so that it can be reproducible.
Improvement in description of Result section so that it can be better interpritable.

Reviewer 2 ·

Basic reporting

Figure 2A and 2B - Dear authors, increase font size for easier viewing and reading.

Figure 3A and 3B - PPI network images containing some dozens of nodes and interactions, although didactic, are difficult to visualize and analyze.I strongly suggest increasing the font size of the gene name as well as transferring the image from main article to supplementary material in .PDF format.

Figure 4A and 4B - The font size of the x and y axes should be increased to facilitate visualization and reading for both the four centrality measures and the values.

Line 106 - Please, cite DAVID and version
Huang DW, Sherman BT, Lempicki RA. Systematic and integrative analysis of large gene lists using DAVID Bioinformatics Resources. Nature Protoc. 2009;4(1):44-57.

Line 115 - Please, cite STRING and version
Szklarczyk D, Gable AL, Lyon D, Junge A, Wyder S, Huerta-Cepas J, Simonovic M, Doncheva NT, Morris JH, Bork P, Jensen LJ, von Mering C. STRING v11: protein-protein association networks with increased coverage, supporting functional discovery in genome-wide experimental datasets. Nucleic Acids Res. 2019 Jan; 47:D607-613.D

Line 118 - Please, cite Cytoscape and version
Shannon P, Markiel A, Ozier O, Baliga NS, Wang JT, Ramage D, Amin N, Schwikowski B, Ideker T. Cytoscape: a software environment for integrated models of biomolecular interaction networks Genome Research 2003 Nov; 13(11):2498-504

Line 119 - Please, cite MCODE and version
BADER, Gary D.; HOGUE, Christopher WV. An automated method for finding molecular complexes in large protein interaction networks. BMC bioinformatics, v. 4, n. 1, p. 2, 2003.

Line 127 - Please, cite Network Analyzer and version
ASSENOV, Yassen et al. Computing topological parameters of biological networks. Bioinformatics, v. 24, n. 2, p. 282-284, 2007.

Line 128 - Please, cite CytoNCA and version
Yu Tang, Min Li, Jianxin Wang, Yi Pan, Fang-Xiang Wu. CytoNCA: a cytoscape plugin for centrality analysis and evaluation of biological networks. BioSystems, 2014, DOI: 10.1016/j.biosystems.2014.11.005

Line 142 - Please, cite GEPIA and version
TANG, Zefang et al. GEPIA: a web server for cancer and normal gene expression profiling and interactive analyses. Nucleic acids research, v. 45, n. W1, p. W98-W102, 2017.

Line 150 - Caros autores, creio que o valor de "585 DEGs..." deve ser substituido por "685 DEGs..."

Line 150 - Dear authors, the value of "585 DEGss ..." should be replaced by "685 DEGs ..."

Experimental design

Dear authors, the PPI network was constructed based on the differentially expressed genes identified in previous experiments, after the four centrality measures were calculated.According to Dilluca (2017), in bacteria, the degree of interaction has correlation with essential proteins, an assertion supported by other articles including experimental ones. However, this statement is true when the PPI network is constructed with all the genes of the organism. Thus, a sub-network of differentially expressed genes does not reflect the dynamics and the biological condition of the organism, consequently generates bias in the measures of centrality.

The STRING version database has 5.879.727 interactions for Homo sapiens, 420.534 having score >= 0.7, and all must be used to calculate the measures of centrality, even if only DEGs genes are described in the results. I also suggest uses score >= 0.7 from STRING because it reflects very reliable interactions.

Reference: DILUCCA, Maddalena; CIMINI, Giulio; GIANSANTI, Andrea. Topological transition in bacterial protein-protein interaction networks ruled by gene conservation, essentiality and function. arXiv preprint arXiv:1708.02299, 2017.

Validity of the findings

Dear authors, the enrichment analysis was performed for 685 DEGs, however, the final result concentrates on seven out of eight genes identified. Thus, I suggest transferring the "GO and KEGG pathway enrichment analysis" step to the end of the "Materials and Methods", analyzing seven target genes found in previous steps, allowing the reader to know in which these seven genes stand out in comparison to the other DEGs .

Line 192 and Figura 4A - The degree interaction of biological networks tends to present a power-law distribution, whose graph can be generated directly in the Network Analyzer plugin of Cytoscape. With a partial network of 685 DEGs many interactions were disregarded, probably generating heavy-tailed distribution graph.

Additional comments

The article uses systems biology techniques to correlate data of gene expression, annotation and interaction networks in order to identify possible relevant genes in the article context.

The main contribution of an Bioinformatics work is to reduce the sample field and to suggest genes for future experimental works. Thus, the article fulfills its role, however it follows some suggestions to improve the work.

·

Basic reporting

Language is generally OK. However, another check will be better
Line 105-106: which is an online program that can provides should be "provide".

Experimental design

Line 109-110: Sorting was conducted according to the P-value from small to large, and the top 25 GO terms and the KEGG pathways were selected.

This is not the ideal way. You should not choose or sort based on p-value only. The p-values just give a significance level (p<0.05 and p<0.01 doesn't tell the latter is better). A better way is to calculate the enrichment factor. Also, check if the genes in identified GO terms are common or specific.

Line 115: The STRING database (http://string.embl.de/) provides the method which can construct a PPI network of DEGs.
The string database can construct a network of any given gene list. The authors have created a network of DEGS. Perhaps a rephrasing of the sentence would be better.

Line 172: Module analysis of the NSCLC related network
This analysis needs to be improved.
How many genes are there in different modules that are related to enriched GO terms and pathways?
What is the role of the modules in the NSCLC carcinogenesis and patient's survival?

Centrality analysis of PPI network.
Only four centralities have been calculated. This section can be improved as well.
1. What are the centralities that have the highest resolution?
2. What is the rank intercorrelation among centralities?
3. It is also not clear how the final selection (16) was arrived at?

Line 194: 16 molecules? (or DEGs?) with high topological features were obtained with high degree, betweenness, closeness, and eigenvector

Validity of the findings

It has to be clearly stated what is the novelty of the work, and how will it be helpful to other researchers.
The findings can also be validated through the search of the literature where others might have reported some of the genes for involvement in NSCLC.

---

## Round 0.2 · accepted · Accept

Your article is Accepted.

·

Basic reporting

Authors have implemented most of the suggestions and may be accepted now for publication

Experimental design

OK

Validity of the findings

OK

Additional comments

Authors have implemented most of the suggestions and may be accepted now for publication

Reviewer 2 ·

Basic reporting

The authors accepted and executed the recommendations of the first review, or justified the reasons.

Experimental design

The authors accepted and executed the recommendations of the first review, or justified the reasons.

Validity of the findings

The authors accepted and executed the recommendations of the first review, or justified the reasons.